Aquatic insects dealing with dehydration: do desiccation resistance traits differ in species with contrasting habitat preferences?

Pallarés Susana susana.pallares@um.es 1
Velasco Josefa 1
Millán Andrés 1
Bilton David T. 2
Arribas Paula 3 4 5
1 Department of Ecology and Hydrology, Universidad de Murcia , Murcia , Spain
2 Marine Biology and Ecology Research Centre, School of Marine Science and Engineering, University of Plymouth , Plymouth , United Kingdom
3 Department of Life Sciences, Natural History Museum London , London , United Kingdom
4 Department of Life Sciences, Imperial College London , London , United Kingdom
5 Island Ecology and Evolution Research Group, IPNA-CSIC , La Laguna, Tenerife , Spain
Sánchez Marta
Electronic publication date: 2016 Aug 31
Publication date: 2016
Volume: 4
Electronic Location ID: e2382
Received 2016 May 30; Accepted 2016 Jul 30
Copyright: ©2016 Pallarés et al.
Copyright year: 2016
Copyright holder: Pallarés et al.
License: This is an open access article distributed under the terms of the Creative Commons Attribution License, which permits unrestricted use, distribution, reproduction and adaptation in any medium and for any purpose provided that it is properly attributed. For attribution, the original author(s), title, publication source (PeerJ) and either DOI or URL of the article must be cited.
License URL: https://creativecommons.org/licenses/by/4.0/

Keywords: Water content, Coleoptera, Lotic, Water loss, Cuticle, Salinity, Enochrus, Lentic, Inland salt water, Drought

Funding: I+D+i project CGL2013-48950-C2-2-P FEDER funds University of Murcia Royal Society UK Spanish Ministry of Economy and Competitiveness This research was part of the I+D+i project CGL2013-48950-C2-2-P (Spanish Ministry of Economy and Competitiveness) cofinanced with FEDER funds, and also supported by a PhD grant from University of Murcia to S.P and two post-doctoral grants from the Royal Society UK (Newton International Program) and from the Spanish Ministry of Economy and Competitiveness (Juan de la Cierva Formación Program) to PA. The funders had no role in study design, data collection and analysis, decision to publish, or preparation of the manuscript.

==============================
Background

Desiccation resistance shapes the distribution of terrestrial insects at multiple spatial scales. However, responses to drying stress have been poorly studied in aquatic groups, despite their potential role in constraining their distribution and diversification, particularly in arid and semi-arid regions.

Methods

We examined desiccation resistance in adults of four congeneric water beetle species (Enochrus, family Hydrophilidae) with contrasting habitat specificity (lentic vs. lotic systems and different salinity optima from fresh- to hypersaline waters). We measured survival, recovery capacity and key traits related to desiccation resistance (fresh mass, % water content, % cuticle content and water loss rate) under controlled exposure to desiccation, and explored their variability within and between species.

Results

Meso- and hypersaline species were more resistant to desiccation than freshwater and hyposaline ones, showing significantly lower water loss rates and higher water content. No clear patterns in desiccation resistance traits were observed between lotic and lentic species. Intraspecifically, water loss rate was positively related to specimens’ initial % water content, but not to fresh mass or % cuticle content, suggesting that the dynamic mechanism controlling water loss is mainly regulated by the amount of body water available.

Discussion

Our results support previous hypotheses suggesting that the evolution of desiccation resistance is associated with the colonization of saline habitats by aquatic beetles. The interespecific patterns observed in Enochrus also suggest that freshwater species may be more vulnerable than saline ones to drought intensification expected under climate change in semi-arid regions such as the Mediterranean Basin.

Introduction

Maintaining water balance is fundamental for organismal survival, small animals such as insects being especially vulnerable to dehydration (Addo-Bediako, Chown & Gaston, 2001). Desiccation resistance therefore shapes the distribution of insect species at multiple spatial scales, both ecologically and biogeographically (Kellermann et al., 2009), and will determine the way insect taxa respond to increased temperatures and greater seasonal fluctuations in water availability in the face of climate change (Chown, Sørensen & Terblanche, 2011).

The role of desiccation resistance for insect vulnerability could be particularly important in arid and semiarid regions such as the Mediterranean Basin. In these areas, many lowland inland waters show spatial and temporal flow intermittency, because they are subjected to intense summer droughts (Hershkovitz & Gasith, 2013; Millán et al., 2011). During the dry period, some small and shallow lentic waterbodies can remain completely dry for months. In intermittent streams, flow connectivity is disrupted; some reaches dry out while others can retain water in receding pools. The predicted increase of the duration and frequency of droughts in Mediterranean-climate regions in the context of global change threatens the persistence of many of their endemic aquatic organisms (Filipe, Lawrence & Bonada, 2013; IPCC, 2013; Lawrence et al., 2010).

However, a large knowledge gap still exists in the way aquatic macroinvertebrates respond to droughts (Robson, Chester & Austin, 2011). In the case of aquatic insects, most studies are focused on desiccation-resistant eggs or dormant larvae stages (e.g., Benoit, 2010; Juliano et al., 2002; Woods & Singer , 2001), but little is known on the effects of dryness on species which lack these resistant stages (e.g., most aquatic beetles, Strachan, Chester & Robson, 2015). Although some species of water beetles resist the dry phase of temporary waters in microrefuges in situ, as adult or larvae stages (e.g., Davy-Bowker, 2002; Stubbington et al., 2016), winged adults of many species show a more resilient strategy, dispersing by flying from drying sites to more favourable wet habitats (Bilton, 2014; Strachan, Chester & Robson, 2015). The duration of exposure to drying stress during dispersal depends on specific biological traits (e.g., flight ability) and landscape configuration, i.e., the availability of suitable habitats that may serve as wet refuges and the distance and connectivity between them (Datry et al., 2016; Larned et al., 2010). But even short exposures to drying stress are challenging for flying aquatic insects, because flight activity is associated to a strong dehydration (Dudley, 2000).

Studies of geographical variation in responses to desiccation stress in terrestrial insects have typically demonstrated that species and populations from xeric environments show a greater ability to deal with dehydration than those from mesic areas (e.g., Chown, 1993; Gibbs & Matzkin, 2001; Le Lagadec, Chown & Scholtz, 1998; Schultz, Quinlan & Hadley, 1992). Different responses to desiccation have also been related to patterns of habitat and microhabitat choice in arthropods (e.g., Chown, 1993; De Vito et al., 2004; Gereben, 1995; Lapinski & Tschapka, 2014), including aquatic species (e.g., Wissinger, Brown & Jannot, 2003; Yoder et al., 2015). In general, these studies show that some physiological mechanisms linked to drying stress such as the control of water loss rate have an important plastic and adaptive component, whilst others, such as the tolerance of water loss, are less variable across species (Chown, Le Lagadec & Scholtz, 1999; Hoffmann & Harshman, 1999). In inland water ecosystems, even congeneric macroinvertebrate species show remarkable differences in the strategies and particular traits to deal with desiccation, and such variability is strongly associated with the frequency and duration of droughts in their habitats (Strachan, Chester & Robson, 2015).

In Mediterranean inland waters, a number of genera of water beetles belonging to different families contain species which are adapted to different parts of the fresh—hypersaline gradient (Millán et al., 2011). Organisms dealing with osmotic stress in saline waters face similar physiological challenges to those imposed by desiccation; i.e., maintaining water balance and compensating for the increase in the osmotic concentration of internal fluids (Bradley, 2009). In light of this, interspecific differences in desiccation resistance may correlate with salinity tolerance, so that species inhabiting saline waters are expected to be potentially more resistant to desiccation than those in lower salinity ranges (Arribas et al., 2014). In addition, species living in relatively short-lived small lentic water bodies, which are more unstable systems over evolutionary and ecological time-scales than lotic systems (see Ribera, 2008 for details), may also be expected to have higher desiccation resistance compared to related lotic taxa. These predictable differences have already been found between lotic and lentic congeneric beetle species in other traits such as dispersal capacity (Arribas et al., 2012), behavioural responses to acute thermal stress (Pallarés et al., 2012) and salinity tolerance (Céspedes et al., 2013).

Together with behavioural responses (e.g., use of microrefuges, burrowing) or aestivation, invertebrates have evolved a variety of physiological adaptations to cope with desiccation stress (Hershkovitz & Gasith, 2013; Strachan, Chester & Robson, 2015). These fall under two main strategies (Chown & Nicolson, 2004; Edney 1977): (1) avoiding desiccation through the reduction of water loss and increases in body water content (desiccation resistance, e.g., Gray & Bradley, 2005) and (2) withstanding the loss of a significant proportion of body water (desiccation tolerance, e.g., Benoit et al., 2007; Suemoto, Kawai & Imabayashi, 2004). In insects, mechanisms regulating cuticle permeability are the major component of desiccation resistance because the cuticle represents their main avenue for water loss (Benoit et al., 2010; Chown & Nicolson, 2004; Hadley, 1994). Cuticle permeability is related with the cuticle thickness (Crowson, 1981; Harrison et al., 2012; Reidenbach et al., 2014), but can be actively regulated through changes in the amount and composition of surface lipids (Gibbs & Rajpurohit, 2010; Stinziano et al., 2015). Water loss has shown to be non-linear following exposure to desiccation in a range of taxa (e.g., Benoit et al., 2007; Arlian & Staiger, 1979). Greater water loss rates occur during initial hours of exposure and decrease as body water content approaches lethal levels, suggesting that water loss is actively regulated by dynamic mechanisms. As a consequence, the water content of an individual at a particular moment could influence its water loss dynamics and ultimately its survival under drying stress. On the other hand, body size can affect desiccation resistance in arthropods in a number of ways. In general, larger body mass allows a higher proportion of water and lipid content (Lighton, Quinlan & Feener jr, 1994; Prange & Pinshow, 1994), and additionally smaller insects may show higher mass-specific water loss rates due to higher surface area–volume ratios (Chown, Pistorius & Scholtz, 1998; Schmidt-Nielsen, 1984; Williams & Bradley, 1998). Such effects of size on water loss rates have been seen both inter- (e.g., Chown & Klok, 2003; Le Lagadec, Chown & Scholtz, 1998) and intraspecifically (e.g., Renault & Coray, 2004).

Here we compared responses to desiccation stress in flying adults of four aquatic beetle species of the genus Enochrus. These species are specialists of either lentic or lotic waters of differing salinity, providing an ideal model to explore the potential relationship between specific desiccation resistance traits of aquatic insects and the main inland waters habitat types. We measured their survival and recovery ability following controlled exposure to drying stress and explored key traits related to desiccation resistance to: (i) determine whether congeneric species with different habitat preferences differ in desiccation resistance and (ii) explore the extent to which inter-individual differences in water loss rates are shaped by body size, cuticle thickness and/or water content in these insects. We predicted that species from most saline habitats would show higher desiccation resistance than less saline tolerant ones. Additionally, species living in lentic waters could have evolved a higher resistance to desiccation than lotic ones.

Material and Methods

Study species

The genus Enochrus (Coleoptera: Hydrophilidae) has representatives living across a wide variety of inland waters of differing salinities (from 0.5 g/L to >80 g/L in the study area). The four species used in this study show different salinity ranges and optima both in laboratory (Pallarés et al., 2015) and nature (Arribas et al., 2014): E. halophilus (Bedel, 1878) (fresh-subsaline waters), E. politus (Kuster, 1849) (hyposaline), E. bicolor (Fabricius, 1792) (mesosaline) and E. jesusarribasi Arribas and Millán, 2013 (hypersaline). All species live in shallow water close to the margins of occupied water bodies, but differ in their habitat preferences across the lentic-lotic divide, being found in lentic (E. halophilus and E. bicolor) and lotic waters (E. politus and E. jesusarribasi) (see Table 1 for more detailed habitat information). These species do not show any resistant form to face desiccation in situ at any stage of their life cycle. Therefore, their main strategy to deal with droughts in temporal and/or intermittent systems relies on the dispersal ability of adults, which move from drying to wet sites. These movements may occur between different or within the same waterbodies, depending on the landscape configuration and habitat availability (Datry et al., 2016; Larned et al., 2010).

Table 1 Habitat parameters of Enochrus species, together with collection sites.

Species	Habitat preferences	Collection sites	
	Conductivity range a (mS cm−1)	Conductivity optimum b (mS cm−1)	Habitat type	Locality	Latitude	Longitude	
E. halophilus	0.47–23.00	6.25 (subsaline)	Temporary-lentic	Pétrola pond, Albacete	38.8471	−1.5589	
E. politus	1.50–133.40	19.32 (hyposaline)	Intermittent-lotic	Chícamo stream, Murcia	38.2175	−1.0511	
E. bicolor	2.10–86.00	34.96 (mesosaline)	Temporary-lentic	Mojón Blanco pond, Albacete	38.8002	−1.4301	
E. jesusarribasi	14.90–160.00	62.14 (hypersaline)	Intermittent-lotic	Rambla Salada stream, Murcia	38.1263	−1.1182	
Notes.

a Field conductivity data were obtained from Biodiversity database of the Aquatic Ecology Research Group, University of Murcia.

b Ranges of conductivity of each category (mS cm−1): Freshwater: <1, Subsaline: 1–10, Hyposaline: 10–30, Mesosaline: 30–60, Hypersaline: >60 (Montes & Martino, 1987).

Experimental procedures

Adult specimens (approx. 50 per species) were collected from different localities all in southeastern Spain and representing the optima salinity conditions of each species (see Table 1). For logistic reasons, we used specimens from one single location per species. Such locations were selected minimizing distances between each other and so have a comparable climatic regime. All species were collected during the spring 2014, therefore the specimens used for the experiments were all mature adults from the winter generation and presumably had not been previously exposed to desiccation stress in natural conditions.

Specimens were maintained for 4–7 days in the laboratory at 20 ± 1°C in aerated tanks with water from collection sites (i.e., at the same salinity of their habitat) and fed with macrophytes also collected in the source localities. For comparative purposes, insects were kept 48 h before desiccation experiments in a dilute medium (ca. 0.1 mS cm−1) at 20 ± 1°C and 12:12 light:day cycle in a climatic chamber (SANYO MLR-351; Sanyo Electric Co., Ltd., Moriguchi City, Osaka, Japan), without access to food. The medium was prepared by dissolving the appropriate amount of marine salt (Ocean Fish; Prodac, Cittadella, Italy) in distilled water.

The experimental protocol and variables recorded in controlled desiccation experiments are showed in Fig. S1. For each specimen studied we obtained the initial fresh mass (M0) as a surrogate of size, initial water content (WC0; % wet mass to initial fresh mass), cuticle content as a surrogate of cuticle thickness (CC; % of cuticle mass to initial fresh mass), water loss rates (WLR; % of water lost to initial fresh mass per unit time) and total water loss after the corresponding treatment (WL; % of water loss to total water content). For this, groups of 20–25 individuals of each species were dried on blotting paper, weighed on a balance accurate to 0.01 mg and placed individually into clean 15 mL open glass vials. These were kept for 6 h in a glass desiccator containing silica gel (Sigma-Aldrich, Madrid, Spain) at 20 ± 1°C. Relative humidity, monitored with a hygrometer (OM-EL-USB-2-LCD; Omega Engineering, Seville, Spain), dropped from approx 40% (laboratory humidity) to 20 ± 5% within the first 2 h and remained stable within this range until the end of the trial. The experimental conditions were optimized trough pilot trials in order to detect differences among species, within their tolerance limits and in a reasonable experimental time (to avoid additional stress such as starvation). The remaining specimens (N = 10–20 individuals per species) were used as a control under no desiccation stress. They were kept in glass vials placed in a closed tank with deionized water in the base, producing a relative humidity ≥90%. After 6 h, surviving specimens from control and test groups were re-weighed for estimation of water loss rates and allowed to recover for 24 h in 100 mL containers with 40 mL of the dilute solution. Some studies have shown that rehydration may result in an excessive increase in specimens’ water content (overhydration stress, e.g., Lopez-Martinez et al., 2009; Yoder et al., 2015). However, we checked in pilot trials (data not shown) that the species here studied recovered their initial water content after rehydration, with no significant water gains. Mortality was monitored after desiccation exposure and after the recovery period. Specimens were then dried at 50°C for 48 h and re-weighed for estimation of the initial water content. A subgroup of 20 individuals per species from the test group were also immersed in 4 mL of 2M NaOH(aq.) for 48 h at room temperature to allow tissue digestion, rinsed in distilled water, dried and weighed again for estimation of cuticle content (Harrison et al., 2012). Specimens were sexed after the experiment by examining genitalia under a Leica M165C stereomicroscope.

Data analyses

Interspecific comparison of desiccation traits

Fresh mass, water loss rate, water content and cuticle content were compared among species using generalized linear models (GLMs) with species as factor, followed by Bonferroni post-hoc tests. Gaussian error distribution and identity link function were used for fresh mass, water content and cuticle content models; and gamma distribution for water loss rate which showed a positively skewed distribution. To account for the potential effects of sex and body size in desiccation resistance, sex and the interaction of sex and species were included as predictors, as well as fresh mass in comparisons of water loss rate, water content and cuticle content (e.g., Addo-Bediako, Chown & Gaston, 2001; Terblanche et al., 2005). Model residuals were checked for normality and homoscedasticity assumptions.

Relationships between desiccation resistance traits within species

To determine the possible effects of initial water content, cuticle content and size (fresh mass) on inter-individual variation in water loss rate, the relationship between water loss rate and each variable was explored for each species separately using GLMs. Gaussian error distribution and identity link function were used when data met a normal distribution. When this assumption was not met, different link functions (log) or different error distributions (Gamma) were implemented, and the model with the lowest AIC was selected.

All the statistical analyses were carried out using R v. 3.0.1 (R Core Team 2015).

Results

Interspecific comparison of desiccation traits

The water and cuticle contents of the four studied species ranged from 60 to 68% and 12–23% M0, respectively. Mean water loss rates of specimens exposed to desiccation ranged from 2.22 to 3.57% M0 h−1, with a total water loss after 6 h of desiccation exposure of 19.3 –39.1% WC0. Specimens in the control group showed very little water loss (approx. 0.5% M0 h−1 anda maximum water loss of 6% WC0) (see Table S1 for species comparative data).

All desiccation resistance traits differed significantly between species (Table 2). Females showed higher fresh mass and water content than males in all species (see sex and sex ×species effects in Table 2). Despite significant interspecific differences in mean fresh mass (Fig. 1A), the effect of initial body mass on the other trait comparisons was not significant (Table 2).

Table 2 GLM results on interspecific differences in fresh mass (M0), water loss rate (WLR), water content (WC0) and cuticle content (CC) across Enochrus species (N = 20 per species).

Trait	Predictors	df	F-value/ χ2a (Explained deviance)b	P	
M0 (mg)	Sp	3	37.627	<0.001	
	Sex	1	14.206	<0.001	
	Sp × Sex	3	0.607	0.613	
			(0.651)		
WLR (% M0 h-1)	Sp	3	2.718	<0.001	
	M0	1	0.126	0.161	
	Sex	1	0.004	0.799	
	Sp × Sex	3	0.007	0.990	
			(0.397)		
WC0 (% M0)	Sp	3	22.086	<0.001	
	M0	1	1.387	0.243	
	Sex	1	4.736	0.033	
	Sp ×Sex		0.335	0.800	
			(0.519)		
CC (% M0)	Sp	3	27.019	<0.001	
	M0	1	3.067	0.085	
	Sex	1	0.027	0.870	
	Sp × Sex	3	1.629	0.192	
			(0.593)		
Notes.

a F-value for GLMs with gaussian distribution (M0, WC and CC); χ2 for GLMs with gamma distribution (WLR).

b (null deviance—residual deviance/null deviance).

Figure 1 Interspecific comparison of desiccation resistance traits in Enochrus species.

Letters below the boxes indicate significant differences between species (Bonferroni post-hoc tests, P < 0.05). Boxplots represent Q25, median and Q75, whiskers are Q10 and Q90 and dots are outliers.

The species living in fresh–subsaline waters (E. halophilus) showed a significantly higher water loss rate, but this did not differ significantly amongst the other three species (Fig. 1B). Initial water content was higher in the meso and hypersaline species (E. bicolor and E. jesusarribasi) than in the subsaline and hyposaline ones (E. halophilus and E. politus) (Fig. 1C). The species showed similar cuticle contents, except for E. halophilus which had the highest value (Fig. 1D). No consistent patterns between lotic and lentic species were observed for any of the measured traits.

No mortality occurred during exposure to desiccation (except for one specimen of E. halophilus). Enochrus halophilus showed a limited capacity to recover after desiccation (44% of the tested specimens died during the recovery period vs. only one specimen in each of the other species). The observed mortality can be mainly attributed to desiccation stress because 100% survival occurred in the control group in all species.

Relationships between desiccation resistance traits within species

In general, the desiccation resistance traits showed high inter-individual variability in all species studied (see Figs. 1 and 2). A significant positive relationship was found between individual water loss rates and water content in all species except for E. halophilus (Fig. 2A). In contrast, cuticle content was not related to water loss rate in any species (Fig. 2B), and these were also independent of initial body mass (Fig. 2C).

Figure 2 Relationships between individual water loss rates (WLR) and initial water content (WC0), cuticle content (CC) and fresh mass (M0) for Enochrus species.

P-values and deviance (D2) are showed for the statistically significant relationships (P < 0.05).

Discussion

On the basis of our investigations, desiccation resistance in Enochrus water beetles appears to be associated with habitat salinity, but does not differ between species occupying lotic and lentic water bodies. The more saline-tolerant species studied (E. bicolor, E. jesusarribasi and E. politus) showed lower water loss rates than the freshwater-subsaline species (E. halophilus). Furthermore, within these three saline species, the meso and hypersaline ones (E. bicolor and E. jesusarribasi) had significantly higher initial water content than the hyposaline E. politus. Indeed, these interespecific differences in water control efficiency seem to be relevant in terms of survival under drying stress, as E. halophilus was also the most sensitive species to the conditions tested here. In consequence, assuming that the species may tolerate similar levels of water loss (Chown, Le Lagadec & Scholtz, 1999 ; Hoffmann & Harshman, 1999), the studied saline tolerant species showed a clear physiological advantage over freshwater ones under desiccation conditions.

Arribas et al. (2014) suggested that salinity tolerance in water beetles could be based on a co-opted mechanism originally developed for desiccation resistance, relying on the temporal correlation of global aridification events and the phylogenetic ages of saline lineages. The pattern found here of stronger desiccation resistance in aquatic species living in saline waters is clearly consistent with this hypothesis and emphasizes the important role that traits associated with coping with osmotic and desiccation stress could have in shaping the ecological diversification of Enochrus. Also, in line with the relationship between desiccation and salinity tolerance seen across the beetles studied here, intraspecific studies of corixid populations found similar responses to the two stressors (e.g., Cannings, 1981), and salinity acclimation was showed to confer desiccation resistance in an Antarctic midge (Elnitsky et al., 2009). Salinity imposes similar stress on aquatic organisms as that resulting from desiccation during air exposure at the cellular level (i.e., water loss and increase of the osmotic pressure) (Evans, 2008; Bradley, 2009). In consequence, shared genetic and physiological mechanisms might underlie resistance to these two factors, as found with other related stressors such as desiccation and cold (e.g., Everatt et al., 2014; Holmstrup, Hedlund & Boriss, 2002; Levis, Yi & Lee, 2012).

Our study found no direct association between desiccation resistance and the lotic/lentic habitat divide. Previous studies on water beetles have shown that lentic taxa have a higher colonization ability (i.e., the ability of a species to disperse and establish new populations) than lotic related species, resulting in larger geographical ranges and lower population genetic structure (Abellán, Millán & Ribera, 2009; Hof et al., 2012; Ribera, 2008). Dispersal capacity and thermal tolerance seem to be the main traits driving this lotic/lentic pattern in water beetles (e.g., Hjalmarsson, Bergsten & Monaghan, 2015) and particularly in two of the species here studied, E. jesusarribasi and E. bicolor (Arribas et al., 2012; Pallarés et al., 2012). The two lotic species studied here are restricted to the Iberian Peninsula and Morocco whilst the lentic ones are distributed across larger areas, including northern Europe (Millán et al., 2014), but no clear patterns in desiccation resistance traits were found accordingly. Therefore, desiccation resistance could play a secondary role to differences in dispersal capacity in shaping the colonization ability of water beetles. In this point it should be noted that Enochrus species’ occurrence across different habitat types will be also constrained by the limited desiccation resistance of eggs and larvae, being the latter likely the most desiccation-sensitive stage because of their thinner cuticles. In addition, desiccation resistance might show inter-population variability (e.g., Hoffmann & Harshman, 1999) as a result of physiological plasticity or local adaptations. Despite our study on adults from populations on similar climatic regimes but different habitats allows for a robust comparison across species, further studies on multiple stages and populations are needed to deeply understand the relationship between habitat occupation and resistance to desiccation in this group.

Beetles are one of the groups of arthropods best adapted to desiccation, with species from desert or semi-desert areas typically representing the extremes in tolerance to dehydration. For example, the terrestrial spider beetle Mezium affine shows daily water losses as little as 0.3% per day and the ability to survive up to three months with no food or water (Benoit et al., 2005). Surprisingly, the highest tolerance to water loss (89% of the body water content) has been reported for a fully aquatic beetle, the haliplid Peltodytes muticus (Arlian & Staiger, 1979). Since they occupy the shallow margins of waterbodies, Enochrus species may be expected to be intermediate in desiccation resistance between strictly terrestrial beetles and those occupying deeper water such as many diving beetles (Dytiscidae) (Beament, 1961; Holdgate, 1956; Wigglesworth, 1945). However, it is difficult to establish a comparative framework because of the few existent data on desiccation resistance traits in adult aquatic insects and the multiple and contrasting approaches and/or experimental conditions used to measure them. The water contents of the four Enochrus species (60%–68% of fresh mass) were consistent with the typical 62% of most beetles (Hadley, 1994). Water loss rates, ranging from 2.2 to 3.6% of initial mass at 20% RH, appear to be comparable to those reported for the extraordinary desiccation resistant P. muticus, which lost ca. 5.4% of initial mass per hour under more severe conditions (0% RH) (Arlian & Staiger, 1979). Nevertheless, the total water losses that the studied species reached after the desiccation treatment (Table S2) were close to the limit of dehydration tolerance of most insects (20–30% of water content) (Hadley, 1994). Although such water loss was measured under an unrealistic humidity in natural conditions (20% RH), a combination of high temperatures (>30°C) and low humidity (40%–50% RH) is frequent in the natural habitats of these species. Prolonged exposures to such conditions in nature may result in extremely stressful conditions and high mortalities of local populations of the studied species, but further research is needed to identify desiccation level and duration thresholds under natural conditions for each particular species.

The analysis of traits at the individual level is essential for further exploration of the mechanisms regulating water loss rate. In Enochrus species, water loss rates were positively related to the specimens’ initial water content. These relationships were relatively weak (D2 = 0.4–0.6) due to high inter-individual variation in both traits, which might be associated to age, sex or the physiological state of the individuals (e.g., Chown, Le Lagadec & Scholtz, 1999; Lyons et al., 2014; Matzkin, Watts & Markow, 2007). Despite this variability, resistance to water loss seems to be partly a function of individual water content, as beetles with a higher initial proportion of water lost it faster than those with lower values. This suggests that a critical level of water loss may induce active mechanisms for water conservation (e.g., changes in cuticular permeability), which might be “relaxed” when organismal water content rises above this threshold. Such regulation is concordant with the nonlinearity of water loss following exposure to desiccation found in many fully terrestrial insects (e.g., Arlian & Staiger, 1979; Benoit et al., 2007).

Although we used cuticle content as a potential surrogate of cuticle permeability, since increased cuticle thickness is associated with desiccation resistance in insects adapted to arid conditions (Crowson, 1981; Elias, 2010), this trait showed no relationship with water loss rates in any Enochrus species. In addition, in interspecific comparisons, the species with the highest mean water loss rate had the highest cuticle content. A recent study also showed that cuticle thickness in adult mosquitoes appeared not to affect desiccation resistance (Reidenbach et al., 2014). Therefore, the validity of cuticle thickness as proxy for cuticular permeability could be very different across taxa and may perhaps have low resolution for intra-generic comparisons. In some terrestrial insects, changes in the composition and quantity of cuticular hydrocarbons appear to be the main mechanism through which they can modulate cuticular permeability (e.g., Hadley, 1978; Nelson & Lee Jr, 2004; Stinziano et al., 2015; Toolson, 1982). In aquatic insects, similar mechanisms may shape responses to dehydration occurring both in exposure to air or hyperosmotic aquatic medium but to date even basic cuticular properties in such taxa have received little study (but see Alarie, Joly & Dennie 1998 for an example).

Despite the fact that many previous studies suggest that body size affects water loss rate in arthropods (e.g., Chown, Pistorius & Scholtz, 1998; Lighton, Quinlan & Feener jr, 1994; Prange & Pinshow, 1994) our results suggest that both interspecific and inter-individual size differences do not significantly affect desiccation resistance in these water beetles. Although large size (lower area-to-volume ratio) might be expected to be beneficial for survival under desiccating conditions (Chown, Pistorius & Scholtz, 1998; Schmidt-Nielsen, 1984), important trade-offs could arise as a result of increases in body size (Chown & Gaston, 2010; Chown & Klok, 2003). This could be particularly true in the case of aquatic insects living in fluctuating or temporary waters, such as the beetles studied here, where rapid larvae development and small body size are common, alongside other r-selected traits (Millán et al., 2011; Williams, 1985).

Conclusions

This study is the first to explore both interspecific and inter-individual variation in desiccation resistance traits within a group of closely related aquatic insects. Our results suggest that control of both water loss rate and water content may be key mechanisms for dealing with desiccation stress in adult water beetles and suggest an association between salinity tolerance and desiccation resistance. Further studies are required to evaluate the ecological and evolutionary consequences of interspecific variation in key desiccation resistance traits, but our results point to habitat-mediated differences (saline vs. freshwater) in the vulnerability of water beetle species to a higher frequency and intensity of droughts expected in semi-arid regions.

Supplemental Information

Figure S1 Experimental procedure and variables measured in desiccation experiments

Click here for additional data file.

Table S1 Summary of variation in desiccation resistance traits in control and treatment groups of Enochrus species

Click here for additional data file.

Data S1 Raw data obtained in desiccation experiments and variables used for statistical analyses

Click here for additional data file.

We thank F. Picazo, S. Guareschi and all members of the research group Aquatic Ecology of the University of Murcia for helping with beetle species field collection and M. Botella-Cruz for helping with experimental procedures. The manuscript was greatly improved by the suggestions of Tomas Ditrich and two other anonymous reviewers.

Additional Information and Declarations

Competing Interests

Author Contributions

Data Availability

The authors declare that there are no competing interests.

Susana Pallarés conceived and designed the experiments, performed the experiments, analyzed the data, contributed reagents/materials/analysis tools, wrote the paper, prepared figures and/or tables.

Josefa Velasco and Andrés Millán conceived and designed the experiments, contributed reagents/materials/analysis tools, reviewed drafts of the paper, field sampling.

David T. Bilton conceived and designed the experiments, contributed reagents/materials/analysis tools, reviewed drafts of the paper.

Paula Arribas conceived and designed the experiments, analyzed the data, reviewed drafts of the paper.

The following information was supplied regarding data availability:

The raw data has been supplied as Data S1.

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
