# Peer review of "Aquatic insects dealing with dehydration: do desiccation resistance traits differ in species with contrasting habitat preferences?"

_PeerJ, doi:10.7717/peerj.2382_

## Round 0.1 · original submission · Minor Revisions

Dear Authors

We have received reviews from 3 referees who appreciate the significance of your study and agree that only minor revision is needed to make your ms suitable for publication. I ask you to pay special attention to comments about the experimental design and whether experiments are reasonably representative of natural conditions. Please, integrate the requested information in the method and discussion sections.

Reviewer 1 ·

Basic reporting

Acceptable, no comments.

Experimental design

Acceptable, no comments.

Validity of the findings

Acceptable, no comments.

Additional comments

Overall

This study investigates dehydration in aquatic insects, specifically in relation to salinity and water flow. The study is interesting and well conducted. I have a few minor comments.

Specific comments:

1. There are a few studies that should be cited and discussed. Specifically, Benoit et al, 2005 (Journal of Insect Physiology) assessed dehydration in beetles and Yoder et al. 2015 (Journal of Experimental Biology) assessed rate of water flow on water balance in caddisflies. These could provide some interesting points for discussion.
2. Line 145: The sentence on protected species is confusing.
3. Line 212-215: The recovery period might be a different stress (rehydration stress, Lopez-Martinez et al. 2009 Journal of Comparative Physiology B) and should be discussed. This means the drying and recovery is dehydration and rehydration stress (two separate stresses).
4. Line 352: Capitalize ecology.

·

Basic reporting

The manuscript meets standards of PeerJ and general scientific literature. I have only minor comment to the Fig. 2 - font size (species names) differs among subfigures A, B, C.

Experimental design

The experimental design is appropriate for the research, I have only few remarks. The authors should just add more details about the species and collecting sites – the text should answer these questions:
a) Do some of the species occur also in temporary sites? If yes, does it leave to other site or aestivate?
b) Were the collecting sites temporary or permanent? Authors just provide general information on unstable lentic water bodies (l. 93 – 94), but not describe the collecting sites from this point (there is just note that these beetles live in fluctuating or temporary waters – l. 316).
c) What date and phase of the life cycle were the beetles collected? There is mentioned (l. 286-287) that inter-individual variation in the measured parameters may be due to the physiological state of the individuals, but the information on passed important life history events affecting the physiological state of individuals (e.g. diapause, mating, oviposition, dispersal/migration etc.) are missing.

Validity of the findings

Again, all results are well presented and discussed. I have just one suggestion - as the water loss was probably not linear during the desiccation treatment, I suggest reporting the water loss rate not as % M0 per hour, but as total water loss per whole desiccation treatment (6 hours).

Reviewer 3 ·

Basic reporting

No comments

Experimental design

No comments

Validity of the findings

No comments

Additional comments

General comments:
The authors studied the desiccation resistance of adult aquatic beetles that are found across a salinity gradient in Spain’s freshwater environments. Understanding whether some aquatic invertebrates are more resistant or vulnerable to drying stress will be important for how organisms may respond to more frequent or unpredictable drying predicted by climate change. As the authors point out, while there are an extensive number of studies on terrestrial invertebrate responses to dehydration, there are surprisingly few for aquatic invertebrates. Therefore, this well-designed study is a great addition to further our understanding of how well adapted aquatic invertebrates may be to drying and salinity stresses.

There were several main points that should be addressed:

1. The laboratory experiments conducted in this study were well executed and analyzed appropriately. However, how realistic are the short-term laboratory drying conditions compared to the natural environment? It would be good to include more details about how frequent and the typical duration of drying these four species experience in the natural environment. Was the relative humidity during the laboratory conditions (20%) consistent with natural drying conditions?

2. I would presume that when the lentic and lotic habitats dry they remain dry for longer periods than the laboratory conditions (6 hours). Would these beetles respond differently (e.g. survival, water content, etc.) under longer periods of desiccation? Other studies have included longer duration desiccation experiments using larval aquatic insects (Wissinger et al. 2003, Galatowitsch and McIntosh 2016) and might be worth including in the discussion.

3. The four Enochrus species were each sourced from one representative habitat. By sourcing individuals from one location might not capture the potential variability in desiccation resistance among populations. Please address how sourcing each species from single representative habitats might limit your laboratory results to broader natural patterns. Also for these source locations had the beetles been previously exposed to natural drying conditions during larval development or as early adults? If some species had been exposed to drying prior to the experiment there might have been prior selection for the characteristics observed in the laboratory. Please address these points in the methods under “study species” and in the discussion.


Specific comments:

Introduction
L57: Please replace the “and” between “seasonal droughts” and “high” with “along with”. This will help reduce the awkward number of “and” in the sentence.
L60-63: Awkward sentence. Please make more concise or break into multiple sentences.
L65: “drying stress” may be more appropriate than “desiccating conditions”.
L68-72: Please break this sentence into two. I suggest a full stop after Robson et al. 2011, and starting the following sentence with “In”.
L71: Have most studies of aquatic insects focused on dormant larval stages?
L73-76: Consider rearranging this sentence, with “species and populations from arid environments” following “report”.
L84: Is “frequency of habitat drying” more appropriate than “temporality of their habitats”?
L94: Please clarify what is meant by “geological and ecological time-scales”
L99: The sentence starts with “behavioural responses”, but these have not been addressed in the previous paragraphs beyond a brief mention of dispersal capacity. Please explain these behavioural responses or remove from the beginning of the paragraph.
L99-105: This is a long awkward sentence. Consider using numbers for the two main strategies. For example replace “one primarily” with “1)” and later “2)” before “withstanding”.
L107-110: Confusing sentence, please consider rewriting.
L126: “main habitat divisions in aquatic insect lineages” is confusing. Please clarify.

Materials and methods
L141-142: To be consistent with the previous citations please include Arribas and Millan, 2013 in parentheses.
L144: To be consistent with the previous use of lotic and lentic earlier in the manuscript please replace “standing” with “lentic” and “running” with “lotic”.
L145-146: Something seems to be missing from this sentence.
L149-151: Was the optimal salinity from the source habitat maintained during the initial 4-7 days in the laboratory? Also, did the macrophytes serve as food or substrate?

Discussion
In the introduction and discussion you mention that the adult beetles are known to disperse to avoid desiccation. Did any of the adults exhibit flight behaviour during the desiccation trials?
L238: Consider adding “phylogenetic” before “ages”
L245-248: Please explain how “aerial desiccation” is different from “desiccation” at the end of this sentence.
L254: Include a space after Hjalmarsson et al., before 2015.
L260-263: Confusing sentence, consider breaking into multiple sentences.
L263: By “along the life cycle” are you referring to larvae and eggs? If so, are these more aquatic stages likely more vulnerable to salinity and desiccation? Adults have thicker cuticles and presumably breathe air from the water surface. Would this make adults less vulnerable to these stresses?
L298: “Interespecific” should be “interspecific”.
L305-308: As mentioned in comments for L263, could the cuticular properties in the beetle larvae be even more important for their response aerial desiccation and osmotic stress than the adults?

References
L360-362: The beginning of each word in the title should not be capitalized.
L373-375: Italicise “Cenocorixa bifida hungerfordi”.
L418: Italicise “Nebria”
L421: Italicise “Drosophila”
L451-453: The beginning of each word in the title should not be capitalized.
L517-518: The beginning of each word in the title should not be capitalized.

---

## Round 0.2 · accepted · Accept

Dear Authors,

I appreciate your detailed revision, which addressed all reviewers’ comments and made several point clearer including relevant additional information and analysis. Based on this revised version, I’m happy to accept it for publication in PeerJ.

Congratulations!